# Chagas Disease Infection Reactivation after Heart Transplant

**DOI:** 10.3390/tropicalmed5030106

**Published:** 2020-06-29

**Authors:** Maria da Consolação Vieira Moreira, José Renan Cunha-Melo

**Affiliations:** 1School of Medicine, Federal University of Minas Gerais (UFMG), Av. Alfredo Balena 190, Belo Horizonte CEP 30130-110, MG, Brazil; mariacvmoreira@gmail.com; 2Department of Surgery, School of Medicine, Federal University of Minas Gerais (UFMG), Av. Alfredo Balena 190, Belo Horizonte CEP 30130-110, MG, Brazil

**Keywords:** chagasic cardiomyopathy, heart transplantation, Chagas disease reactivation, cardiac allograft rejection, treatment of reactivation

## Abstract

Chagas disease, caused by a *Trypanosona cruzi* infection, is one of the main causes of heart failure in Latin America. It was originally a health problem endemic to South America, predominantly affecting residents of poor rural areas. With globalization and increasing migratory flows from these areas to large cities, the immigration of *T. cruzi* chronically-infected people to developed, non-endemic countries has occurred. This issue has emerged as an important consideration for heart transplant professionals. Currently, Chagas patients with end-stage heart failure may need a heart transplantation (HTx). This implies that in post-transplant immunosuppression therapy to avoid rejection in the recipient, there is the possibility of *T. cruzi* infection reactivation, increasing the morbidity and mortality rates. The management of heart transplant recipients due to Chagas disease requires awareness for early recognition and parasitic treatment of *T. cruzi* infection reactivation. This issue poses challenges for heart transplant professionals, especially regarding the differential diagnosis between rejection and reactivation episodes. The aim of this review is to discuss the complexity of the Chagas disease reactivation phenomenon in patients submitted to HTx for end-stage chagasic cardiomyopathy.

## 1. Brief Historical Context

The first human heart transplantation (HTx) was performed by Christian Barnard in December 1967 [1]. The patient was treated for rejection and died 18 days after HTx due to pneumonia and sepsis [2]. The available data today show that, in spite of fantastic scientific and technological developments, the problems of rejection, infection, allograft failure, cardiac allograft vasculopathy (CAV), and Chagas reactivation still constitute major problems faced by heart transplant centers. 

The introduction of new immunosuppressive agents, such as cyclosporine A in 1980 and FK 506 (tacrolimus) in 1989, improved survival rates for all heart transplant recipients and definitively qualified HTx as the treatment of choice for end-stage heart failure [3]. In addition, advances in donor procurement, surgical techniques, post-HTx care, organ preservation, and the big jump in technological development (bringing great positive impacts in anesthesia, surgical interventions, and immunosuppression protocols), in conjunction with the accumulated experience at heart transplant institutions, have contributed to a great decrease in acute rejection episodes and an improvement of the overall results of HTx [3]. However, the donor shortage is still a worldwide problem [4]. To overcome this difficulty, the concept of donation after circulatory death donors and the extended criteria donors have been used as mechanisms to increase the donor pool [5,6]. Currently, not only has the number of advanced heart failure patients risen, but the heart transplant candidates are more complex [5,7]. In spite of the great advances in the field of transplantation, heart transplant professionals and patients still have to face many challenges. Regarding HTx in chagasic patients, the experiences in Brazil in the 1980s established the viability of HTx for Chagas cardiomyopathy as an alternative form of treatment [8,9]. In spite of this, a worldwide spread of Chagas disease and the complexity of reactivation of the *Trypanosoma cruzi* infection have brought major problems to the field. Therefore, more than 50 years later, the differential diagnosis between rejection and *T. cruzi* reactivation poses similar challenges as those faced by Dr. Barnard. 

## 2. The Economic Burden of Chagas Disease

At present, Chagas disease poses a major public health challenge for the Americas, as well as for non-endemic regions around the world, including the United States, Canada, Countries in the European Union, Australia, and Japan [10].

A model for studies of the economic burden of Chagas-disease expansion beyond tropical and sub-tropical zones has been proposed to calculate the healthcare costs and disability-adjusted life-years (DALYs) for individuals, countries, and regions, resulting in an estimate of 7,968,094 infected individuals across the word [10]. 

For each chronic Chagas disease-infected individual, the calculated cost per year has shown an average cost of USD 383, USD 1762, and USD 2162 in Latin America, Europe, and USA, respectively. The calculated annual cost per person is USD 4660 with a lifetime cost per person of USD 27,684. The global costs per year are estimated to be USD 7,190,000,000, and the cost per lifetime around USD 188.8 billion. This summary of economic burden suggests more attention and effort is needed in the control of Chagas disease [10].

## 3. Peculiarities of Chagas Disease in the Heart Transplant Setting

Chagas disease infects nearly seven million people in the world, the majority in Latin America [11]. The clinical course of the disease is characterized by an acute phase with patent parasitemia and proliferation of amastigote forms in several tissues. Symptoms subside in a few months and most patients pass to an asymptomatic form of the chronic phase, named the indeterminate phase, with low parasitemia in the blood and tissues. Decades after an initial infection, 20% to 30% of the patients develop chronic cardiopathy, including arrhythmias, conduction defects, sudden cardiac death, and heart failure [12]. The mortality of heart failure patients due to Chagas disease is higher than that observed in other cardiomyopathies [13]. 

Despite the fact that Chagas disease is a long-life infection, the anti-trypanosomal therapy for infected people during the chronic phase of the disease is not clearly effective and remains a challenge [14]. 

Heart transplantation is a therapeutic option for those patients with advanced heart failure refractory to medical therapy. Reactivation of Chagas disease is a common finding under immunosuppressive conditions, such as AIDS, autoimmune diseases, cancer (and the chemotherapy used to treat it), and obviously, pharmacological immunosuppression to avoid allograph rejection [7,15].

### 3.1. Patient Selection 

The experience in Brazil in the 1980s established the viability of HTx for Chagas cardiomyopathy as an alternative form of treatment. Currently, HTx is an important therapeutic tool for chagasic patients with advanced heart failure and constitutes the third leading indication for HTx in Brazil [8,9].

The indications and contraindications for HTx follow the classic criteria for other etiologies of heart failure, but some peculiarities are often observed [7,8]. Chagasic patients have lower pulmonary artery pressure, pulmonary vascular resistance, and transpulmonary gradient, which can reduce right ventricular dysfunction, a frequent complication in the post-operative period of HTx [8]. Thus, some transplant centers do not perform cardiac manometry by right cardiac catheterization if the systolic pressure in the pulmonary artery, as estimated by the Doppler-echocardiogram, is <50 mmHg [8]. In general, chagasic patients have a less favorable social and cultural profile, which makes the feasibility of complex procedures such as HTx a challenge. However, there does not seem to be a relationship between the socioeconomic situation and the evolution after HTx [16]. The possibility of megaesophagus and megacolon should be evaluated, which, depending on the severity, may constitute contraindications to HTx [17].

Serology for *T. cruzi* infection in all potential donors and recipients from endemic areas is mandatory, and a donor who tests positive is not accepted for heart recipients. Potential organ donors and recipients should always be screened for the possibility of Chagas disease in endemic countries as well as in non-endemic countries, where the potential donor/recipient ratio has a positive epidemiology [12].

### 3.2. Immunosuppression Strategies

One of the goals of transplantation science is to equilibrate the immunosuppression, to prevent rejection, and to reduce infection with the occurrence of drug toxicity. Immunosuppressive regimens to prevent rejection can be done by induction (i.e., intense early perioperative/post-operative techniques) and maintenance (i.e., for life) [18,19,20]. Some categories of recipient candidates (juvenile patients, patients with history of pregnancy or multiple blood transfusions, those on mechanical circulatory support (MCS), and sensitized patients) are supposed to benefit from induction therapy; whereas calcineurin inhibitors, antimetabolite agents, proliferation signal inhibitors, and glucocorticoids are used for maintenance immunosuppression [4]. 

The main inducing agents are polyclonal anti-thymocyte immunoglobulins (polyclonal antibody—thymoglobulin) and interleukin 2 receptor inhibitors (IL-2), such as daclizumab and basiliximab, which have low immunogenicity [4].

Basic immunosuppressive therapy for the maintenance of cardiac transplant patients in general necessarily includes a calcineurin inhibiting agent, namely, cyclosporin A or tacrolimus. These agents must be associated with mycophenolate mofetil (MMF), mycophenolate sodium, azathioprine, rapamycin, or everolimus. Prednisone is associated with this standard regimen, and in most patients, it can be suspended 6 months after transplantation, in the absence of rejection [4]. 

In the context of Chagas disease, induction and/or maintenance immunosuppressive therapy can reactivate the *T. cruzi* infection [8,20]. There are no studies comparing the various immunosuppression regimens in chagasic patients, however, a greater number of reactivations have been diagnosed with the use of MMF [21]. Therefore, it would be recommended that chagasic patients receive immunosuppressive therapy with the lowest possible doses, as long as there is no rejection. To prevent rejection-induced reactivation, strategies based on generic principles have been proposed [8]: Reduction in the immunosuppression (which facilitates graft rejection);The use of low doses of several drugs whenever feasible;The avoidance of excessive doses of immunosuppressive agents.

## 4. Allograph Rejection following Heart Transplantation

Allograft rejection is an important cause of death after heart transplant, in spite of the scientific advances in the field [22,23,24].

The main types of rejection are as follows: hyperacute, acute cellular (ACR), and antibody-mediated (AMR). 

The International Society for Heart and Lung Transplantation (ISHLT) has redefined the pathological diagnosis of both ACR and AMR rejections by grading their severity. ACR is graded as: 0R (no rejection), 1R (mild), 2R (moderate), or 3R (severe). AMR is graded based on immunological (I) or histopathological (p) parameters as follows: pAMR 0 (negative); pAMR1(H+) (p positive and I negative); pAMR1(I+) (p negative and I positive), pAMR2 (both I and p positive), and pAMR3 (severe p) [22,23]. Antibody-mediated rejection has not been frequently reported after HTx in chagasic patients but it can occur [8].

Acute cellular rejection seems to be the predominant type of rejection in such patients. A 70% occurrence of ACR was reported within the first year after HTx, with a 10% mortality rate in chagasic recipients [9,15,25]

The incidence of rejection in chagasic and non-chagasic recipients does not seem to be different [8]. Endomyocardial biopsy, in spite of its invasiveness, is the most used method to monitor and diagnose ACR. Under routine histopathological staining techniques, parasites may not be seen, and the inflammatory infiltrate of rejection (grade 2R or 3R) and reactivation episodes are quite similar; thus, the differential diagnosis between inflammation caused by rejection or reactivation is a difficult task. The protocols for treatment of allograft rejection in both chagasic and non-chagasic recipients are similar. The majority of cases presenting ACR respond properly to pulse corticosteroid therapy, although rescue therapy may be required for selected cases [8]. 

In addition, a high percentage (up to 43%) of inflammatory infiltrates found to be compatible with the diagnosis of 2R or 3R rejection do not respond to immunosuppressive therapy, but show a good response to antitrypanosomal drugs. Therefore, the detection of an inflammatory mononuclear infiltrate in the endomyocardial biopsy is not enough to rule out the diagnosis of Chagas disease reactivation and poses a challenge, as the most common drug to abort rejection (corticosteroid) may facilitate Chagas disease reactivation. Over 85% of patients have at least one rejection episode before reactivation occurs [8,25].

## 5. Reactivation

*T. cruzi* reactivation after HTx is closely related to aggressive immunosuppression. As high immunosuppression protocols induce more frequent Chagas disease reactivation episodes after HTx and affect 20% to 45% of recipients in the first year, an early diagnosis of reactivation is necessarily aimed at pre-emptive therapy. Type I T helper immune response is an important mechanism involved in the *T. cruzi* infection. It is well established that high-dose corticosteroid is able to modify the cytokine profile of type I T helper lymphocytes and is associated with the antiproliferative effect of MMF on T lymphocytes. This environment constitutes a favorable condition for Chagas disease reactivation, which properly treated results in less than 1% mortality. Current evidence indicates that the probability of reactivation could be as high as 90% at 2 years following HTx. Symptoms may be quite similar to those seen in the acute phase of Chagas disease as well as in rejection episodes and include fever, anorexia, myalgia, diarrhea, panniculitis, myocarditis, meningoencephalitis, and encephalic vascular accident [26,27,28,29]. 

Different regimens using immunosuppressive drug associations have not been tested in HTx for chagasic cardiomyopathy. MMF in maintenance immunosuppression seems to be more closely associated with reactivation episodes than the regimens that do not use this drug. Therefore, strategies to change the immunosuppression regimen, such as replacement of MMF by azathioprine or decreasing MMF, have been proposed [21], although no randomized clinical trials are yet available. An early reduction in immunosuppressant agents (especially corticosteroids) is recommended to prevent reactivation, but this approach may facilitate rejection episodes [20,30]. 

It should be remembered that Chagas disease may affect non-chagasic patients receiving organs from donors with chronic Chagas disease [17,31]. 

### 5.1. Reactivation Diagnosis

Reactivation episodes of Chagas disease after HTx have been described as myocarditis, panniculitis, meningoencephalitis, and brain abscess (Figure 1). Myocarditis, the most frequent manifestation, may be asymptomatic or present severe symptoms compatible with heart failure or cardiogenic shock. New skin nodules are characteristic of reactivation. These nodules may ulcerate and the presence of nests of amastigotes is a common skin biopsy finding. Cardiac allograft involvement can manifest as tachycardia, cardiac arrhythmias, A-V blocks, ventricular dysfunction, and cardiogenic shock. Reactivation episodes may induce clinical acute Chagas disease-like symptoms, including fever, anemia, jaundice, liver function test alterations, myocarditis, and neurologic symptoms, secondary to the parasitic effects on the central nervous system. [12,29,32]. However, reactivation episodes may occur without symptoms. 

Asymptomatic individuals, with tissue or blood samples persistently showing the presence of *T. cruzi* are strong candidates for infection reactivation. The Latin American Guideline for the Diagnosis and Treatment of Chagas Heart Disease has listed some risk factors associated with *T. cruzi* infection reactivation as follows: the number of rejection episodes; the presence of malignancy; immunosuppression grade; use of MMF; autoimmune diseases; HIV infection; and other immunosuppression status [8,30].

The diagnosis of reactivation episodes can be made when symptoms of Chagas disease infection are present. However, diagnosis from only the clinical features is insufficient, i.e., both clinical symptoms and the positive detection of parasites in the blood, cerebrospinal fluid, endomyocardial biopsy, or other tissue samples must coexist. The detection of parasites is made by direct examination of the blood or cerebrospinal fluid, or by biopsies of any infection site which show *T. cruzi* forms under conventional, immunohistochemistry, or immunofluorescence techniques [8].

At present, polymerase chain reaction (PCR) analysis of the blood, endomyocardial biopsy (EMB), or other tissue biopsies are reliable methods to confirm the presence of *T. cruzi* as compared to other techniques. However, EMB is an invasive approach and although being considered a safe procedure, when performed by an experienced operator, complications and sequelae, such as: access site hematoma, right ventricular perforation, chordae tendineae damage, right bundle branch block, arrhythmias, tricuspid regurgitation, may occur. In addition, coronary artery-to-right ventricular fistula, permanent tricuspid valve regurgitation and scarring of the right interventricular septum, compromising the amount of retrieval tissue in future biopsies may occur [33]. One advantage of PCR analysis is that the *T. cruzi* detection precedes the clinical manifestations of reactivation by two or more months. As a consequence, the time required to reach a diagnosis is less than that required by standard parasitological methods. The evolution of the PCR method has led to the appearance of a number of PCR variants. For example, quantitative PCR has a 95.7% sensitivity and a 100% specificity for parasite detection in the acute phase of Chagas disease. Currently, PCR diagnosis is a precious tool which guides physicians as to whether patients should begin receiving anti-parasite drugs or changes in the immunosuppression protocol [26,34,35].

It is well recognized that the isolation of *T. cruzi* from the blood of chagasic recipients (xenodiagnosis, blood culture) is not considered pathognomonic for the diagnosis of Chagas disease reactivation. The same tests may be positive in patients with the chronic form of Chagas disease. In addition, the Strout test is also an alternative for reactivation diagnosis. On the other hand, serological tests are useless for reactivation diagnosis. Their main indication in organ transplantation is when seronegative patients receive organs from seropositive donors, with the most frequently used being ELISA, indirect immunofluorescence (IIF), and indirect hemagglutination (IHA) [8,12,30,35]. A tendency to substitute the immunofluorescence assay (IFA) for a new test, named the trypomastigote excreted–secreted antigens (TESA) immunoblot, in some centers in the U.S.A. has been observed [36].

In countries where Chagas disease is not endemic, failure to identify patients with Chagas disease reactivation constitutes a major medical and social problem, as severe or fatal outcomes may supervene the incapacity to establish a proper diagnosis. [37,38]. 

The concept of reactivation must, therefore, be redefined as even if no clinical symptoms are evident, reactivation can be diagnosed by an increase in parasitemia, detected either by direct parasitological techniques or by PCR. In this context, a patient may be considered to be presenting reactivation if a current PCR is positive and the previous PCR result was negative. Similarly, if the former test showed lower parasitemia than the current one, the reactivation diagnosis can be accepted [17,34]. Considering that rejection and reactivation may occur several times, that rejection usually precedes reactivation episodes, that symptoms of each condition are similar, and that both conditions may coexist, a differential diagnosis is fundamental for the treatment. Reactivation may then occur several times even if the first episode was properly treated. Despite the possibility of several reactivation episodes in the transplanted heart, no chagasic chronic cardiomyopathy in the recipient has been described [8], and so the role played by *T. cruzi* itself in the infection reactivation is not clear. 

There exists an obvious need to monitor for *T. cruzi* reactivation in order to allow the start of specific treatment, mainly in patients without clinical symptoms [29,39]. Chagasic recipients must be followed-up after HTx and monitored for *T. cruzi* infection reactivation, much as they are monitored for rejection, not only routinely but also at any time when the presence of clinical suspicion occurs. Variations in the protocol occur depending on the transplantation center policy. One such policy suggests monitoring for reactivation by histological demonstration of amastigotes at 1, 3, 5, 6, 9, and 12 months after HTx, or whenever the presence of signs or symptoms are present, such as new skin lesions, fever, or overt acute myocarditis [8]. However, no scientific definition about when and how the monitoring protocol should be applied is available. Some centers agree that monitoring for reactivation should take place during the same time that biopsies aiming at allograph rejection detection are done; others prefer monitoring for reactivation every week for 2 months after HTx, then every 2 weeks until the sixth month, and then, once a month from 6 to 12 months [40].

Differential diagnosis between acute cellular rejection and chagasic myocarditis due to reactivation episodes is not an easy task and poses difficulties. The histopathological findings in both conditions are represented by the presence of lymphocytes surrounding the cardiac muscle cells. Slight differences between the morphology of the infiltrate may be seen. Histological diagnosis of reactivation is confirmed if tissue nests of amastigote forms or antigens are detected. The possibility of toxoplasmosis should always be discarded, which makes the use of immunohistochemistry mandatory to establish the diagnosis of reactivation; skin nodule biopsies showing lymphocytes and histiocytes accumulation may be the clue for confirmation of Chagas reactivation. Intense proliferation of *T. cruzi* amastigotes inside macrophages and endothelial cells warrant the diagnosis. However, in the absence of amastigote forms, it is often difficult, even for experienced pathologists, to diagnose the etiology of skin lesions. If parasitic nests of amastigotes are not found in histological sequential sections, the diagnosis of reactivation remains presumptive [18,41]. In the presence of parasitic nests, the way to confirm the etiological nature of the nests is to detect *T. cruzi* antigens by immunohistochemistry. PCR analysis helps a lot in this context. If PCR can distinguish between living and dead trypanosomes after reactivation treatment, it is not known, but certainly parasite DNA and antigen may persist for a period of time in lesions and thus a positive PCR may not always be indicative of an active site of infection [29,41]. 

Polymerase chain reaction seems to be the topline laboratory method contributing to the diagnosis of reactivation, but both conditions may coexist, and one or more episodes of rejection usually precede the reactivation process. Surveillance for life by a multidisciplinary team is therefore fundamental to the outcome of HTx. The major challenge is to monitor laboratory evidence of Chagas disease reactivation, allowing for etiological treatment before the onset of clinical manifestations, and to prevent severe symptoms and damage to the transplanted heart. [26,34,35,42]

### 5.2. Results of Heart Transplantation in Chagasic Cardiomyopathy Concerning Reactivation

Evaluation for Chagas reactivation was conducted in 107 adult patients submitted to HTx at InCor, São Paulo, Brazil. The diagnosis of reactivation was accepted only in the case of histological confirmation. In 43 out of the 107 studied patients (40.2%), Chagas reactivation was confirmed. Twenty-three of these 43 patients (53.5%) had the diagnosis confirmed by endomyocardial biopsy, 11 (25.6%) by blood samples testing, 8 (18.6%) by skin, and 1 (2.3%) by brain biopsy. The majority of patients used corticosteroids and MMF. No death or severe graft dysfunction were related to the reactivation episodes [9]. 

A similar study was conducted in the U.S.A. for Chagas disease after HTx in 31 recipients. Evidence of *T. cruzi* infection reactivation was found in 19 (61%). The median time for reactivation was 3 weeks after HTx (from <1 to 89 weeks). PCR monitoring allowed the diagnosis before manifestation of clinical symptoms in 18 patients, and in only one patient reactivation was diagnosed after the onset of clinical manifestation. No death in this reactivation group was reported after a median of 60 weeks [43]. 

The HTx in patients with chagasic cardiomyopathy seems to present a better outcome than in non-chagasic recipients. Several factors have been suggested to contribute to this outcome: lower age, less co-morbidities, lower severity grade of rejections, lower incidence of CAV, lower prevalence of pulmonary hypertension, and no history of cardiac surgery before HTx [8,42]. The reported higher incidence of neoplasms after HTx for Chagas disease has not been confirmed in transplanted patients, possibly because of the use of lower doses of immunosuppressive agents, with the aim of avoiding Chagas disease reactivation. The advent of malignancy in HTx of chagasic recipients contributes to death in about 2% of them [8,42,43,44].

At present, infection and rejection are the major causes of death amongst chagasic recipients of HTx, occurring in 21% and 10–14% of patients, respectively. Cardiac allograft vasculopathy and neoplasms do not seem to be frequent causes of death in HTx chagasic recipients [9]. 

The native heart may present myocarditis secondary to the presence of *T. cruzi* parasites. This finding does not seem to be associated with Chagas disease reactivation after HTx [8]. 

### 5.3. Etiological Treatment of Reactivation 

The etiological treatment of Chagas disease is an attractive approach as it could modify the evolution of the disease. Benznidazole and nifurtimox are the anti-trypanosomal drugs of choice, which have been shown to be effective when administered to patients in the acute phase of the disease [12,45]. However, their efficacy on the chronic phase has been a subject of debate [45,46,47,48].

Benznidazole seems to be effective in children with *T.cruzi* chronic infection (early chronic phase, less than 15 years), as demonstrated in two placebo-controlled trials, that showed cure rates of approximately 60%, on the basis of conversion to negative serologic test results after treatment [49,50] A large prospective randomized multicenter study on Chagas cardiomyopathy, benznidazole treatment was unable to prevent the cardiac clinical progression although have reduced the parasitemia [51]. 

The etiological treatment of adults in the indeterminate phase, or very early signs of cardiac involvement, remains unanswered [52,53]. Antiparasitic treatment is not recommended for patients in the chronic phase with advanced Chagas heart disease, as is the case of the heart transplant candidates, since there is no evidence of benefit [12].

The recommendation of benznidazole as a prophylactic agent for all chagasic patients submitted to HTx is questionable due to the lack of available scientific data to support this practice. Besides this, only a percentage of the recipients develop reactivation, and the toxicity and side effects of the drugs might, in some circumstances, be prohibitive. The current recommendation for chagasic HTx is to start anti-trypanosoma treatment whenever Chagas reactivation is confirmed [17,40,41].

Benznidazole and nifurtimox are the trypanocidal drugs of choice, as both compounds are active against trypanosoma. The therapeutic regimens for reactivation are typically made with benznidazole as the first-line drug and nifurtimox as the second option (i.e., for parasite strains resistant to benznidazole) [40,41,42]. 

Benznidazole tablets contain 100 mg of active substance. The drug is absorbed in the intestine, processed by the liver cytochrome P-450 system, and excreted predominantly in the urine with a half-life of 12 hours. The recommended posology is 2.5–5 mg·kg^−1^ two times a day (bid). The proposed duration of treatment is 60 days, with a possible extension, depending on the case, to 90 days [51]. The more prominent collateral effect is an urticariform rash, occurring in about 30–60% of patients, usually appearing in the first week of treatment and treated with anti-histamine drugs or with low doses of corticosteroids. If fever and lymph node enlargement appear, benznidazole intake should be interrupted. Less common adverse effects include a late peripheral polyneuropathy manifesting as pain and tingling in the legs, anorexia, insomnia, and bone marrow suppression (which is of rare occurrence), which also imply treatment interruption. Another option is the use of nifurtimox (120 mg/tablet, 8–10 mg/kg), but this medicine is not available in Brazil. The capacity of benznidazole (and nifurtimox) to eliminate the circulating parasites in about 2 weeks and to affect the host immune response is well known, due to its cytotoxic effects on T-cells. These considerations reinforce the importance of benznidazole as a treatment of Chagas disease reactivation [12,30,48].

Both benznidazole or nifurtimox have been contraindicated in pregnant women, as well as in patients with hepatic or renal failure.

It should be reinforced that the same recipient may evolve with more than one episode of Chagas reactivation, needing multiple treatments and increasing the possibility of adverse effects. 

## 6. Heart Transplantation Complications and Survival

The clinical outcome, morbidity, and mortality rates after HTx are similar when chagasic and non-chagasic recipients are compared in the early post-operative period. The major complications after HTx are related to graft dysfunction (20%); rejection (grade 2R or 3R) (10–20%); acute kidney failure (up to 70%) [54]; bleeding (10%); non-*T. cruzi* infection, mainly those located in the respiratory tract (20–30%) [21]. The MCS for chagasic patients with severe cardiomyopathy is not common in Brazil, but when used, no difference in survival was noticed, as compared to patients without bridged MCS [25]. Right ventricular insufficiency, in general, improves with time.

The survival rate of HTx in chagasic patients is better than that for HTx in patients with no Chagas disease. A lower incidence of post-transplant cardiac allograph rejection and a low mortality related to reactivation of Chagas disease have been reported [44]. The actual overall survival of HTx patients with Chagas cardiomyopathy at 1, 5, and 12 years is better than the survival of recipients of HTx with either ischemic or idiopathic cardiomyopathy (the first and second causes of HTx in Brazil, respectively). The survival rates of HTx for Chagas disease have been reported as 76%, 71%, and 46%, at 6 months, 1 year, and 10 years, respectively [21,26,44]. 

## 7. Conclusions

Chagas disease (American trypanosomiasis) was originally a health problem endemic to South America, predominantly affecting residents of poor rural areas. The migration of *T. cruzi*-infected individuals to large cities and to developed, non-endemic countries has promoted the worldwide dissemination of Chagas disease. Some of these emigrants are submitted to HTx in the new host non-endemic country, and as a consequence, many reports of reactivation of Chagas disease in HTx patients have been published. 

Chagas disease reactivation may occur as a consequence of immunosuppression from several causes, but most published reports of this condition have focused on HTx recipients. 

The high number of HTx for chagasic cardiomyopathy in Brazil has meant that there is an accumulation of experience in this field, and thus, has brought the attention of international scientific organizations looking for partnerships to build global recommendations for prevention, early diagnosis, and treatment approaches concerning Chagas disease. 

The diagnosis of rejection and/or reactivation in chagasic patients submitted to HTx is not an easy task. PCR seems to be the topline laboratory method, contributing to the differential diagnosis between reactivation and rejection episodes. A complication factor in this issue is the occurrence of one or more episodes of rejection, usually preceding the Chagas disease reactivation process.

Efforts to find laboratory evidence of Chagas disease reactivation, thus allowing therapeutic maneuvers before the onset of clinical manifestations, are intended to prevent severe symptoms and damage to the transplanted heart. 

Benznidazole and nifurtimox, in spite of their adverse effects, are the drugs currently eligible for the treatment of Chagas disease reactivation following HTx. 

The etiological treatment of HTx recipients without clinical manifestations of Chagas reactivation but showing a positive PCR assay for *T. cruzi* deserve further investigation and a multicenter trial.

## Figures and Tables

**Figure 1 tropicalmed-05-00106-f001:**
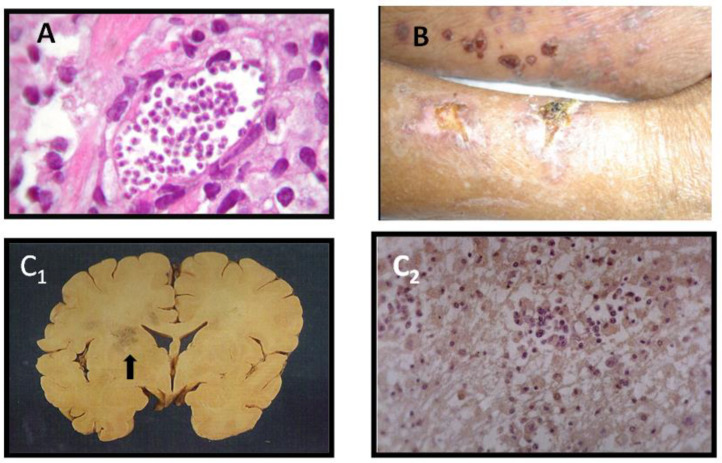
Illustration of Chagas disease reactivation in heart (**A**), skin (**B**), and brain (**C**) in chagasic patients submitted to HTx). (**A**)—Myocarditis in an endomyocardial biopsy showing a nest of amastigotes in the transplanted heart (hematoxylin-eosin staining). (**B**)—Skin lesions in a heart-transplanted chagasic patient. The histology of a biopsied lesion demonstrated a nest of amastigotes (Not shown). (**C**)—Brain lesions in Chagas disease reactivation after heart transplant. (**C1**)—Post-mortem examination showing chagasic encephalitis in a brain macroscopic slice (arrow). (**C2**)—Nests of amastigotes as demonstrated by histopathologic examination (histochemistry: immunoperoxidase technique staining).

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
