# Peer review of "Chagas Disease Infection Reactivation after Heart Transplant"

_tropicalmed, 2020, doi:10.3390/tropicalmed5030106_

Round 1
Reviewer 1 Report
Very interessant "state of the art" about Chagas disease at the time of hart transplantation.
Maybe the beginning about hart transplant should be shortened.
Author Response
Response to Reviewer 1
Dear reviewer, firstly, thank you so much for your praising, guidance and recommendations. Your comments are valuable and very helpful for revising and improving our paper.
Basically the main concern was with the lenght of Introduction. We have gone through the paper and suppressed more 30 (thirty) lines from the Introduction. Furthermore we have done an extense rewriting and rearragement of the wole manuscript. So we do hope this action will match what you have suggested.
Reviewer 2 Report
The topic is very interesting and worth to be published. However, the manuscript is very long and difficult to read. We read several times that it is difficult to distinguish between rejection and reactivation. I agree completely, but at the end of the article I still do not know, what to do. The article is difficult to read. Some sentences are incomplete. Some topics are discussed at different places.
My suggestions
- The manuscript has to be shortened and concentrate only on this difference
- Suggestion have to done, which tests are proposed to at which interval
- How often PCR: If there are different suggestions in the literature, they have to be discussed and finally one should be preferred. I agree that this is difficult for the author, but somebody has to do the decision.
- What to do, if the PCR is positive in the blood or in the tissue. For leishmaniasis we know, that PCR may stay positive in the tissue for years without any clinical relevance, but not in the blood. Under which conditions do you suggest ant parasitic treatment?
- Biopsy of the heart: The authors say that is difficult to distinguish between rejection and reactivation: What is the indication to do it and how often we will have a conclusive interpretation?
- Which are the clinical parameters demanding further examinations.
- A table showing the difference between rejection and reactivation
- Clinical
- Risk factors
- Etc
- Laboratory tests
- Biopsy
- The discussion of prophylactic treatment of benznidazole is well done and relevant for the reader.
- I don’t understand the sentence: antibody mediated rejection has not been documented. What is here the difference to the rejection we see
Author Response
Reply to Reviewer 2
Dear reviewer,
Thank you very much indeed, for your time, guidance and recommendations. Your comments are valuable and very helpful for revising and improving our paper.
1 - The topic is very interesting and worth to be published. However, the manuscript is very long and difficult to read. We read several times that it is difficult to distinguish between rejection and reactivation. I agree completely, but at the end of the article I still do not know, what to do. The article is difficult to read. Some sentences are incomplete. Some topics are discussed at different places.
R- We agree with the comment. To improve the Reading difficulties we have shortened
the introduction, suppressing 30 lines form this section. We also did a major revision in the whole paper and we do think that the manuscript has an improved quality. We have reviewed al the trext, have corrected some inconsistencies and tried to avoid, at the maximum, the repetitions of subjects in this new version. We also tried to improve the sub-heading topics joining similar matters and reviewing the English spelling mistakes.
2- The manuscript has to be shortened and concentrate only on this difference We’ve shortened the manuscript and tried to reinforce the diferences between rejection
and reactivation. We have modified the text in many parts trying to make it more comprehensive.
3- Suggestion have to done, which tests are proposed to at which interval
R- We deliberately decided not to do this because no uniform protocol exists. The impression is that each transplantation center decides based on its belief and experience wuich tests must be done and the interval between tests. We believe that the problems behind this include the agressiveness of EMB and the lack of objective data to establish a correct diagnosis. As the manuscript intend to be a review we decided to avoid giving personal opinions about some of the subjects. A tendency to use less agressive tests for the diagnosis of reactivation exists and was debated, but in our opinion a definitive protocol should be proposed by the scientific organizations.
4- How often PCR: If there are different suggestions in the literature, they have to be discussed and finally one should be preferred. I agree that this is difficult for the author, but somebody has to do the decision.
R- As stated in the previous comment we agree that somebody has to propose theprotocol. However we believe that the oficial representative organizations are the ones legitimate to do so. Also there is a problem concerning the quality of the Medical Institutions doing HTx. In our opinion only very few experienced centers should provide this tytpe of health care. Many hospitals do not have even a PCR machine neither a qualified service of Pathology to process biopsy samples by standard histology, immunohistochemistry or immunofluorescent methods.
5- What to do if the PCR is positive in the blood or in the tissue. For leishmaniasis we know, that PCR may stay positive in the tissue for years without any clinical relevance, but not in the blood. Under which conditions do you suggest anti parasitic treatment?
R- Thank you again for the valuable comment. We believe the kind of comments like yours justify, by themselves, the submission of a manuscript in the field of Chagas disease as we learn so much trying to answer the points raised. We believe that a tissue positive PCR in the heart is patognomonic for reactivation as the donor's heart has no Chagas. So, if you make a EMB and find a positive PCR test this is, in our point of view,enough for the diagnosis of reactivation. The PCR in other tissues of the recipient should be interpreted carefully because the recipient has chronic Chagas disease and might present positivity in a biopsy sample even in the absence of reactivation. We have tried to direct similar points in the manuscript. Treatment should be instituted whenever heart biopsy shows amastigotes or in situations described in the revised manuscript. Parasitic treatment should be done whenever diagnosis of reactivation is made. (see etiological treatment of reactivation)
6- Biopsy of the heart: The authors say that it is difficult to distinguish between
rejection and reactivation: What is the indication to do it and how often we will have a conclusive interpretation?
R- The answer, we believe, is more clear in the revised manuscript. Basically, to be sure about the diagnosis of reactivation is is necessary to detect parasites in the myocardium. However, the presence of parasites in the biopsy does not put away the possibility of a concomitant rejection episode. On the other hand, the absence of parasites in the biopsy is not enough to preclude an episode of reactivation. The matter is really very complex.
7- Which are the clinical parameters demanding further examinations.
R- We think that based only on clinical data is almost impossible to differentiate between rejection and reactivation. If no parasites are found or if the PCR is negative the tendency is to treat for rejection, knowing that both conditions may coexist and also that treatment of rejection episodes may facilitate reactivation episodes. If a patient presents clinical signs of myocarditis, without detectable reactivation, and does not respond to immunossuppressive protocol, the specific treatment for Chagas should be considered.
8- A table showing the difference between rejection and reactivation
Clinical Risk factors Etc
Laboratory tests
Biopsy
R- Thank you for the suggestion. However we find almost impossible to built a table as suggested due to lack of scientific information supporting this difficult task. Chagas reactivation following heart transplant is a very complex medical problem and it is almost impossible to stablish a clearcut difference built in a table. We do hope to improve our understanding about in the future, but, at the moment we do not believe that the suggested table can be built.
9- The discussion of prophylactic treatment of benznidazole is well done and relevant for the reader.
R- Thank you.
10- I don’t understand the sentence: antibody mediated rejection has not been
documented. What is here the difference to the rejection we see
R- Antibody-mediated rejection (AMR) of the cardiac allograft is a poorly defined and challenging diagnosis for transplant recipients and their clinicians. The ISHLT provides 4 categories of diagnostic criteria for AMR : clinical, histopathologic, immunopathologic, and serological assessment. Despite these published criteria, currently >50% of heart transplant centers make the diagnosis of AMR based on cardiac dysfunction and the lack of cellular infiltrates on the heart biopsy. The first description by the ISHLT defined AMR as positive immunofluorescence, vasculitis, or severe edema in the abscence of cellular infiltrate in HTx for non-Chagas cardiomyopathy. Its true incidence is not known, varying from 3% to 85%. Because of the evolving diagnostic criteria and lack of routine screening by most programs, AMR is likely underreported. In published studies describing the incidence of AMR, the diagnostic criteria may include pathological findings, clinical findings, or both. Using such criteria in 870 HTx recipients an incidence of 85% at 100 days was found. In cellular rejection,
clinical descriptors such as recurrent, persistent, or hemodynamic compromise are used to illustrate clinical presentation or clinical severity. AMR is diagnosed in endomyocardial biopsy samples exhibiting complement and immunoglobulin deposits on frozen section, as well as histological changes of endothelial activation and vascular adherence of macrophages, with or without hemorrhage. We believe that the importance of AMR diagnosis is increasing in the HTx settings, and now at least in non-chagasic patients, AMR is associated with allograft failure, decreased survival, increased incidence of CAV, and overall poor prognosis. It may be associated with hemodynamic compromise in up to 50% of patients. After 5 years, 86% of patients with AMR had CAV compared with 22% of control subjects (P<0.001). The incidence of CAV or death in the patients with AMR was twice that of the control subjects (P=0.01) (references 22-24). In transplanted patients for Chagas cardiomyopathy it is not known if reactivation may present clinical and histopathological features that might simulate
some of the findings in AMR.
Reviewer 3 Report
Moreira and Cunha-Melo propose to review Chagas disease infection reactivation following heart transplantation.
This is potentially an important topic not very often addressed. However I got mixed feelings following the reviewing of the manuscript due in part to some lack of clarity and structure of the review and to some typos/English styles. The following considerations when addressed by the authors would enhance greatly the quality of the review in my opinion:
Among others:
- Overall the entire format, structure of the review should be considered to enhance clarity and flow of the review: Chagas disease, role of Heart transplant in CD patients, Issues, etc....
- The abstract does not depict the goal of the review but encloses generalities on Chagas disease; should be completely re-written
- The introduction on the other hand goes straight to CD infection reactivation without short description of the disease itself. should be addressed as the review is targeting a specific part of the disease
- It is hard all along the text to have clarity about the topic discussed: general issues related to Heart transplantation or specific to CD patients; I would suggest to re-write with possibly first commonalities related to Heart transplants in general and then concentrate on CD patients that are the specific goal of the review
- I would have thought that one pre-requisite before Heart Transplantation in CD is treatment with the current Standard of Care, Benznidazole (Bz) and/or nifurtimox (Nfx); the authors should address this more in detail; otherwise re-activation is due to happen following immunosuppression. If it is thought otherwise then should be much described and referenced
- Bz and Nfx are active against all forms of the parasite T. cruzi not only trypomastigotes. should be corrected.
- The authors should also change the statement that these drugs do not cure chronic patients; we just do not have the tools to make this statement / conclusion (time for seroreversion can take decades reason for the search of markers that could give an answer quicker)
- Quite a substantial amount of typos/language and formatting issue (among others, list not exhaustive):
- l. 35: would write "multidisciplinary" rather than mulitproffessional
- l. 57 would write "more importantly for the patients" rather than mainly
- l. 75: Do not understand the meaning of "circulatory death": car accident? vascular system?
- l. 83: format of the bullet points
- l. 179: glucocorticoids
- etc....
Author Response
Reply to Reviewer 3
Dear Reviewer,
Thank you, for your, guidance and recommendations. Your comments are valuable and very helpful for revising and improving our paper.
1. Overall the entire format, structure of the review should be considered to enhance clarity and flow of the review: Chagas disease, role of Heart transplant in CD patients, Issues, etc....
R- We did an extensive revision and have shortened the lenght of the manuscript. We followed your suggestion changing the subtitles to improve the quality and clarity of the paper.
2. The abstract does not depict the goal of the review but encloses generalities on Chagas disease; should be completely re-written.
R- We totally agree. The abstract sent to the periodical was not the one we would like to. We entirely redid the abstract. Thank you.
3. The introduction on the other hand goes straight to CD infection reactivation without short description of the disease itself. should be addressed as the review is targeting a specific part of the disease
R- We have changed the Introduction and cut about 35 lines of it, and added a brief preamble to introduce the matter.
4. It is hard all along the text to have clarity about the topic discussed: general issues related to Heart transplantation or specific to CD patients; I would suggest to re-write with possibly first commonalities related to Heart transplants in general and then concentrate on CD patients that are the specific goal of the review.
R- We modified the approach, attending to your suggestion, and we do think the manuscript is now more readable.
5. I would have thought that one pre-requisite before Heart Transplantation in CD is treatment with the current Standard of Care, Benznidazole (Bz) and/or nifurtimox (Nfx); the authors should address this more in detail; otherwise re-activation is due to happen following immunosuppression. If it is thought otherwise then should be much described and referenced
R- We consider this manuscript a brief review about reactivation. The subject is difficult do be developed as the transplantation centers, in spite of the guidelines, tend to follow their own protocols. The antitrypanosomal drugs used for prevention of reactivation are a matter of great controversies, and the literature is far from being uniform about this topic. We believe this is better addressed in the section treatment of reactivation in the revised edition, but this subject has no unanimity on the literature. .
6. Bz and Nfx are active against all forms of the parasite T. cruzi not only trypomastigotes. should be corrected.
R- This is another subject about which very little has been written. This topic is a matter for discussions being held all over the world, among the expertise authorities in HTx for patients with Chagas cardiomyopathy. Regarding the topic cure, it may be acceppted that the drugs can cure acute forms of the disease and reactivation episodes. However, the discussion about curing the chronic clinical form is still going on. The treatment is able to decrease parasitemia, but as the progression of cardiopathy is not impaired by treatment we think that T.cruzi tissue aggregates may remain, despite the treatment.
7. The authors should also change the statement that these drugs do not cure chronic patients; we just do not have the tools to make this statement / conclusion (time for seroreversion can take decades reason for the search of markers that could give an answer quicker)
R- The statement is not ours, as it can be found in the literature. The treatment results for the chronic form is still far from achieving consensus among the researchers in the field. Both Authors work as Professors of the School of Medicine/UFMG, institution with a long tradition in medical treatment of Chagas disease. The results of treatment with BZ is efficient in the acute phase of Chagas disease but the results of treatment for chronic form of the disease is far from ideal. The repetition of reactivation episodes and their treatment with trypanocidal drugs whenever an episode happens is a kind of indication that might exist a tissue source of parasites not totally erradicated by the available drugs.
8. Quite a substantial amount of typos/language and formatting issue (among others, list not exhaustive):
- 35: would write "multidisciplinary" rather than mulitproffessional
- 57 would write "more importantly for the patients" rather than mainly
- 75: Do not understand the meaning of "circulatory death": car accident? vascular system?
- 83: format of the bullet points
- 179: glucocorticoids
etc....
R- We have done the suggested changes in typos/language and formatting issue. We accept the errors as our exclusive faults. However we sent the manuscript for the English Editing of the Trop Med, as we are not native English speaking, for review. Any way we went throughout the whole manuscript and have made the modifications you have suggested. This step was done for the first submission. Considering this revised version was so deeply modified the Authors have decided to resubmit the revised version to the English Editing section of Trop Med, once more.
Regarding the meaning of “donation after circulatory death (DCD)”: Usually the organs for transplantation are withdrawn from donors with brain death, mechanically ventilated, and anesthethized. DCD is used when the retrieval of organs for transplantation occurs from patients whose death is diagnosis and confirmed using cardiorespiratory criteria. Asystole must be confirmed after five minutes to declare death. To minimize ischemic injury on DCD organs, the cold ischemic time is limited through the use of ex vivo heart perfusion in a warm environment prior to transplantation. Additionally, due to shortage of donors and depending on the need of HTx for a recipient, for higher risk recipients higher risk donors may be accepted. The DCD could be included in this extended criteria donor concept.
Round 2
Reviewer 2 Report
The manuscript is much better and acceptable.
I have only minor comments
Line 191
Chagas reactivation does not only cause encephalitis but also Chagas brain abscess.
Line 204
HIV is not an auto-immundisease
Line 268
Histologic demonstration: Where is the biopsy taken from? Can you please describe in one few words the risks of a heart biopsy?
Line 287
Can PCR distinguish between living trypanosomes causing reactivation and dead trypanosomes after successful treatment? The authors explained in their response this question well. May be this difficulty should be in one sentence.
Do Xenodiagnosis and culture still play a role to distinguish death from living trypanosomes?
Author Response
Response to Reviewers
Reviewer 2
Thank you very much indeed for your time revising our manuscript. Your comments have contributed for additional improvements of the manuscript.
Comments and Suggestions for Authors
The manuscript is much better and acceptable.
I have only minor comments
- Line 191 - Chagas reactivation does not only cause encephalitis but also Chagas brain abscess.
R- We have changed the term encephalitis by the expression “meningoencephalitis, and brain abscess” (line 190).
Line 204 - HIV is not an auto-immundisease
We have change the frase “...; autoimmune diseases, such as HIV infection; and other immunosuppressive status to: autoimune diseases; HIV infection; and other immunosuppression status [8,32].
Line 268
Histologic demonstration: Where is the biopsy taken from? Can you please describe in one few words the risks of a heart biopsy?
R- In this particular paragraph the biopsies were heart biopsies, as the suggestion is to utilize the same biopsy samples to check rejection, to look for histologic Chagas reactivation. We added " during the same time that biopsies aiming at alograph rejection detection are done;
We have described, as suggested, the risks of a EMB. For clarity this information was written along line 222 were biopsy was mentioned in the text and added the following sentence with its reference (lines 222 – 227) : However, EMB is an invasive approach and although being considered a safe procedure, when performed by an experienced operator, complications and sequelae, such as: access site hematoma, right ventricular perforation, chordae tendineae damage, right bundle branch block, arrhythmias, tricuspid regurgitation, may occur. In addition, coronary artery-to-right ventricular fistula, permanent tricuspid valve regurgitation and scarring of the right interventricular septum, compromising the amount of retrieval tissue in future biopsies are among the chronic sequelae [33].
Line 287
Can PCR distinguish between living trypanosomes causing reactivation and dead trypanosomes after successful treatment? The authors explained in their response this question well. May be this difficulty should be in one sentence.
R- We added a brief sentence at line 285 of the revised version, concerning PCR in dead and live trypanosomes:
If PCR can distinguish between living and dead trypanosomes after reactivation treatment, it is not known but certainly parasite DNA and antigen may persist for a period of time in lesions and thus a positive PCR may not always be indicative of an active site of infection [31].
Do Xenodiagnosis and culture still play a role to distinguish death from living trypanosomes?
We believe it is impossible to have a dead trypanosomes giving a positive Xeno or growing in culture. In addition, both Xeno diagnosis and culture have high rates of false negative and may take a long time to get the results (up to three months in some cases).
Reviewer 3 Report
The revised version of the manuscript led to substantial improvements and improved quality.
A few minor points should be considered before final acceptation of the manuscript:
Line 16: "the immigration of chronic T. cruzi hosts to developed, non"; what is a chronic T. cruzi host? should be reformulated with something like "T. cruzi chronically-infected people" or similar
Line 319: "However, the treatment is less effective in the chronic form of Chagas disease [17,19,50,51]. In chagasic patients with cardiomyopathy, benznidazole treatment...". The chronic phase of the disease includes both indeterminate and symptomatic stages. I would clarify here. Moreover, we do not really know if antiparasitic treatment is less effective in the chronic form (i.e. including indeterminate phase) as we have no means in adults to measure efficacy. Moreover the BENEFIT trial was very controversial. therefore I would be much more differentiated in the statement here.
Line 328: "Benznidazole and nifurtimox are the trypanocidal drugs of choice, as both compounds are active against trypomastigotes. The efficacy varies with the geographical region where the reactivation occurs. These differences might be explained by geographical variation of T. cruzi strains" . these 2 drugs are not only active against trypomastigotes but also intracellular amastigotes that are the replicative forms. please correct as this is misleading. During reactivation, there are not only trypomastigotes in blood but also high replication of the parasite in tissues all over.
The statement on variable efficacy depending on the regions should also be more differentiated. See comment above; since we have no real means in the clinic to measure efficacy of treatment it is hard to state that treatment in region X is more efficacious that in region Y irrespective of the variation of T. cruzi strains. Moreover very often mixed infections occur with different strains.
Author Response
Response to Reviewers
Reviewer 3
Thank you very much ofr your precious comments about the manuscript. We have processed the suggestions made and we do believe the manuscript has na improved quality now.
The revised version of the manuscript led to substantial improvements and improved quality.
A few minor points should be considered before final acceptation of the manuscript:
1- Line 16: "the immigration of chronic T. cruzi hosts to developed, non"; what is a chronic T. cruzi host? should be reformulated with something like "T. cruzi chronically-infected people" or similar
R- We followed your suggestion and have modified the sentence as suggested.
2- Line 319: "However, the treatment is less effective in the chronic form of Chagas disease [17,19,50,51]. In chagasic patients with cardiomyopathy, benznidazole treatment...". The chronic phase of the disease includes both indeterminate and symptomatic stages. I would clarify here. Moreover, we do not really know if antiparasitic treatment is less effective in the chronic form (i.e. including indeterminate phase) as we have no means in adults to measure efficacy. Moreover the BENEFIT trial was very controversial. therefore I would be much more differentiated in the statement here.
R- We totally agree with your statement that chronic phase of the disease includes both the indeterminate form and symptomatic stage. We have modidified the text from lines 319 to 323, of the section 5.3 with a more critical analysis and adding 4 new references in this last revised edition. We believe the reading is now more clear and more complete. The modification goes from lines 324 to 336 of the revised version.
3- Line 328: "Benznidazole and nifurtimox are the trypanocidal drugs of choice, as both compounds are active against trypomastigotes. The efficacy varies with the geographical region where the reactivation occurs. These differences might be explained by geographical variation of T. cruzi strains" these 2 drugs are not only active against trypomastigotes but also intracellular amastigotes that are the replicative forms. please correct as this is misleading. During reactivation, there are not only trypomastigotes in blood but also high replication of the parasite in tissues all over.
R- We have decided to suppress the controversy about efficacy of benznidazole against trypomastigotes and amastigotes forms from the text to avoid discussion of a matter that is still quite confusing in the literature, where a few publications say that the effect of trypanocidal drugs is less effective in the chronic form when compared to the acute form or reactivation of Chagas disease episodes. Furthermore the treatment of the chronic form is marginal to the central focus of the manuscript, that is reactivation.
4- The statement on variable efficacy depending on the regions should also be more differentiated. See comment above; since we have no real means in the clinic to measure efficacy of treatment it is hard to state that treatment in region X is more efficacious that in region Y irrespective of the variation of T. cruzi strains. Moreover very often mixed infections occur with different strains.
R- We agree with the Reviewer statement. Cure assessment is controversial because of the lack of specific tests to document parasitological cure. In spite of this it seems to be well accepted that benznidazole is active in the acute stage of Chagas disease (up to 80%), and in early chronic infections, but of limited efficacy against established chronic-stage disease. Regarding the regional variations of T. cruzi strains, according to Zingales et al. Quoted by Sosa-Estani & Segura (Mem. Inst. Oswaldo Cruz 2015; 110(3), 289-298), the T. cruzi itself is genetically diverse. It is divided into six discrete typing units (DTUs), TcI-TcVI, with different prevalence according to the geographic region. DTUs TcI are prevalent in domestic settings, which have a higher risk for vector transmission, in northern South America and Central America, while DTUs TcII, V and VI are prevalent in southern South America (Zingales B, Miles MA, Campbell DA, Tibayrenc M, Macedo AM, Teixeira MM, Schijman AG, Llewellyn MS, Lages-Silva E, Machado CR, Andrade SG, Sturm NR 2012. The revised Trypanosoma cruzi subspecific nomenclature: rationale, epidemiological relevance and research applications. Infect Genet Evol 12: 240-253).
We have suppressed the discussion about the regional variation of T. cruzi strains from the text.